# Diversity of Pectobacteriaceae Species in Potato Growing Regions in Northern Morocco

**DOI:** 10.3390/microorganisms8060895

**Published:** 2020-06-13

**Authors:** Saïd Oulghazi, Mohieddine Moumni, Slimane Khayi, Kévin Robic, Sohaib Sarfraz, Céline Lopez-Roques, Céline Vandecasteele, Denis Faure

**Affiliations:** 1Department of Biology, Faculty of Sciences, Moulay Ismaïl University, 50000 Meknes, Morocco; s.oulghazi@yahoo.fr (S.O.); mmoumni02@yahoo.fr (M.M.); 2Institute for Integrative Biology of the Cell (I2BC), Université Paris-Saclay, CEA, CNRS, 91198 Gif-sur-Yvette, France; Kevin.ROBIC@i2bc.paris-saclay.fr; 3Biotechnology Research Unit, CRRA-Rabat, National Institut for Agricultural Research (INRA), 10101 Rabat, Morocco; slimane.khayi@inra.ma; 4National Federation of Seed Potato Growers (FN3PT-RD3PT), 75008 Paris, France; 5Department of Plant Pathology, University of Agriculture Faisalabad Sub-Campus Depalpur, 38000 Okara, Pakistan; sohaib002@gmail.com; 6INRA, US 1426, GeT-PlaGe, Genotoul, 31320 Castanet-Tolosan, France; celine.lopez-roques@inra.fr (C.L.-R.); celine.vandecasteele@inra.fr (C.V.)

**Keywords:** Pectobacterium, Dickeya, plant pathogen, potato tuber, genome, field sampling

## Abstract

Dickeya and Pectobacterium pathogens are causative agents of several diseases that affect many crops worldwide. This work investigated the species diversity of these pathogens in Morocco, where Dickeya pathogens have only been isolated from potato fields recently. To this end, samplings were conducted in three major potato growing areas over a three-year period (2015–2017). Pathogens were characterized by sequence determination of both the *gapA* gene marker and genomes using Illumina and Oxford Nanopore technologies. We isolated 119 pathogens belonging to *P. versatile* (19%), *P. carotovorum* (3%), *P. polaris* (5%), *P. brasiliense* (56%) and *D. dianthicola* (17%). Their taxonomic assignation was confirmed by draft genome analyses of 10 representative strains of the collected species. *D. dianthicola* were isolated from a unique area where a wide species diversity of pectinolytic pathogens was observed. In tuber rotting assays, *D. dianthicola* isolates were more aggressive than Pectobacterium isolates. The complete genome sequence of *D. dianthicola* LAR.16.03.LID was obtained and compared with other *D. dianthicola* genomes from public databases. Overall, this study highlighted the ecological context from which some Dickeya and Pectobacterium species emerged in Morocco, and reported the first complete genome of a *D. dianthicola* strain isolated in Morocco that will be suitable for further epidemiological studies.

## 1. Introduction

Pectinolytic Pectobacterium and Dickeya *spp.* are causative agents of severe diseases in a wide range of plants of high economic value [1,2]. On potato tubers and stems, the diseases caused by pectinolytic pathogens are soft rot and blackleg, respectively. These pathogens produce a large set of extracellular enzymes that degrade the plant cell wall, resulting in plant tissue decay and maceration. This rotting process causes losses in the production of potato tubers sold both on the food market and as certified seed tubers [3]. The pathogens may be acquired by the host plants from soil and/or from contaminated seed tubers [4]. On plants, pathogen populations remain at a low level in asymptomatic plant tissues, and may become particularly destructive when environmental conditions favor their proliferation and the expression of virulence factors.

*P. atrosepticum* was considered as the primary pathogen responsible for the rotting of stored potato tubers and wilting of potato plants under temperate climates [4]. Other Pectobacterium species frequently associated with damage of potato crops are *P. carotovorum*, *P. brasiliense, P. parmentieri* and *P. polaris* [5,6,7,8,9]. Some Pectobacterium species have also been characterized in some specific areas, such as *P. peruviense* strains isolated from tubers in Peru, and *P. punjabense* species from symptomatic potato plants in Pakistan [10,11]. *P. odoriferum* exhibits a very wide host range, including potato plants [12], while some other species were characterized by a more restricted host range, at least in the fields. Thus, *P. wasabiae* was isolated from symptomatic Japanese horseradish [13]; *P. betavasculorum* was reported almost exclusively on sugar beet [14]; *P. aroidearum* exhibits a preference for some monocotyledonous plants [15]; *P. zantedeschiae* strains were isolated from *Zantedeschia spp.* (Calla lily) [16]; and *P. actinidiae* from symptomatic *Actinidia chinensis* (kiwi fruit) [17]. Recently, some other species, isolated from surface waters, have also been described: *P. fontis, P. aquaticum* and *P. versatile* [7,18,19]. Altogether, 16 *Pectobacterium* species have been described so far [7].

A limited number of Dickeya species, i.e., *D. dianthicola, D. dadantii* and *D. solani*, have been associated with symptoms on potatoes. *D. dianthicola* was first reported on potatoes in the Netherlands in the 1970s, and has been detected since then in many other European countries [20]. *D. dadantii* causes soft rot disease in several members of the *Solanaceae* family, including the potato [21]. Another virulent species, namely *D. solani,* spread rapidly throughout Western Europe [22] and in Russia [23], and into other countries such as Turkey [24], Georgia [25] and Brazil [26]. During the past decade, the taxonomy of Dickeya and Pectobacterium species was revisited following genomic studies bearing on international culture collections and diverse ecosystems around the world [27,28]. By now, the genus *Dickeya* encompasses 10 species: *D. aquatica, D. chrysanthemi, D. dadantii, D. dianthicola, D. fangzhongdai, D. lacustris, D. paradisiaca, D. solani*, *D. undicola,* and *D. zeae* [22,29,30,31,32,33,34]. Bacteria belonging to this genus cause plant diseases in temperate, tropical and subtropical climates [35].

The unambiguous identification of Dickeya and Pectobacterium species is crucial for epidemiological purposes, to develop appropriate prophylactic approaches and quality controls in national and international trade exchanges. Multi-Locus Sequence Analysis (MLSA) provides relevant information for a better understanding of speciation, and hence for proposing pertinent taxa delineations [15,36]. MLSA may exploit gene sequences, obtained by PCR-sequencing of several loci or by whole genome sequencing. Among the loci commonly included in MLSA, the *rrs* sequence is poorly informative at a species level, while the *gapA* gene appeared as an appropriate marker to discriminate the different Dickeya and Pectobacterium species [10,18,19,31,32,37]. Taxonomy of *Dickeya* and *Pectobacterium* gained precision and robustness with additional genome analyses, such as average nucleotide identity (ANI) and in silico DNA–DNA hybridization (*is*DDH) [38]. Comparative genomics is also used to identify species-specific DNA regions. Analysis of the functions encoded by these DNA regions allows the prediction of species-specific metabolic traits. This knowledge contributes to the understanding of both the taxonomy and ecology of the *Dickeya* and *Pectobacterium* pathogens [22,30,31,32,39].

*P. atrosepticum*, *P. carotovorum* and *P. brasiliense* were described in Morocco as the main causative agents of blackleg and soft rot diseases in potato crop [40,41,42,43]. In 2016, *D. dianthicola* was described for the first time in the North of Morocco [44]. In this respect, the main objectives of this study were: (i) to investigate the species composition of the Moroccan *Dickeya* and *Pectobacterium* populations, collected between 2015 and 2017 from diseased potato tubers and stems, (ii) to compare the aggressiveness of some identified pathogens belonging to different species, and (iii) to propose a complete genome of the emerging pathogen *D. dianthicola* in Morocco, that could be used for further studies as a reference genome. This work represents the most important sampling effort of the Pectobacterium and Dickeya potato pathogens in Morocco over the past decade.

## 2. Materials and Methods

### 2.1. Sampling and Isolation of Pectinolytic Bacteria

In 2015, 2016 and 2017, blackleg symptoms were searched for in potato fields in four regions (Meknes, Guigo, Boumia and Larache) in Northern Morocco. Pectinolytic bacteria were isolated from symptomatic plant tissues using crystal violet pectate (CVP) medium as described previously [45]. The CVP plates were incubated at 28 °C for 3 days and colonies that had formed pits were re-streaked onto Tryptone (5 g/L) yeast extract (3 g/L) agar medium (TY). The purified isolates were spotted again on CVP to confirm the pectinolytic activity. The obtained cultures from single colonies were stored in 25% glycerol at −80 °C.

### 2.2. Molecular Characterization of Pectobacterium and Dickeya Isolates

The primer couples Y1/Y2 and ADE1/ADE2 (Appendix A) were used for the identification of isolates belonging to Pectobacterium and Dickeya genera [46,47]. The reaction was carried out in a final volume of 25 μL, containing 1 μL of bacterial DNA (50 ng/µL), 2.5 μL of PCR buffer (10×), 2 μL of MgCl2 (25 mM), 2.5 µL of dNTPs (1 mM), 1U Taq polymerase and 1 µL of each primer (1 µM) and water. The temperature settings for PCR were the same as described before [46,47]. The analysis of PCR products was done by electrophoresis on 2% agarose gels, using PCR products of *P. atrosepticum* CFBP1526^T^ and *D. solani* IPO2222^T^ as control along with the 1 Kb DNA ladder.

Positive strains for either Y1/Y2 or ADE1/ADE2 PCR were further characterized using the *gapA* barcode procedure [37]. All the *gapA* PCR products obtained with gapA-F/gapA-R primers (Appendix A) were sequenced using Sanger technology (GATC Biotech, Konstanz, Germany). The sequences were trimmed using the CLC genomic workbench (V10.1.1, Aarhus, Denmark) and aligned using ClustalW. The phylogenetic analysis of the *gapA* gene was performed as follows: the evolutionary distances were computed using the maximum composite likelihood method (Mega7 software, Pennsylvania State University Park, PA, USA) with 1000 bootstrap. The obtained sequences were deposited in GenBank (Appendix A).

### 2.3. Genome Sequencing

A total of 10 strains representing 5 species identified with *gapA* sequencing were selected for genome sequencing (Table 1). DNA of the 10 isolates (listed in Table 1) was extracted from overnight cultures in TY medium using the MasterPure™ Complete DNA and RNA Purification Kit (Epicenter, Madison, WI, USA) followed by an ethanol precipitation. The quantity and quality control of the DNA was completed using a NanoDrop (Wilmington, DE, USA) device and 1.0% agarose gel electrophoresis.

Paired-end libraries (500 bp in insert size) were constructed for each strain, and DNA sequencing was performed by Illumina NextSeq technology. Sequencing of the library was carried out using the 2 × 75 bp paired-end read module. Illumina sequencing was performed at the I2BC sequencing platform (Gif-sur-Yvette, France).

In the case of the *D. dianthicola* strain LAR.16.03.LID, Nanopore sequencing was also performed. Library preparation and sequencing were performed at the GeT-PlaGe core facility, INRA Toulouse, using the “1D Native barcoding genomic DNA kit (EXP-NBD103 and SQK-LSK108)”, according to the manufacturer’s instructions. At each step, DNA was quantified using the Qubit dsDNA HS Assay Kit (Life Technologies). DNA purity was tested using a NanoDrop device (Thermofisher, Waltham, MA, USA) and size distribution and degradation was assessed using the fragment analyzer (AATI) High Sensitivity DNA Fragment Analysis Kit. Purification steps were performed using AMPure XP beads (Beckman Coulter). Quantities of 5 µg of each DNA (five samples) were purified then sheared at 20 kb using the Megaruptor1 system (Diagenode, Seraing, Belgium). A DNA damage repair step was performed on 3 µg of sample. Then END-repair and dA-tailing of double stranded DNA fragments were performed on 1 µg of each sample. Then, a specific index was ligated to each sample. The library was generated by an equimolar pooling of these barcoded samples. Then adapters were ligated to the library. The library was loaded on a R9.4.1 flowcell and sequenced on MinION instrument at 0.15 pmol within 48 h.

### 2.4. Genome Assembly

Assembly of the Illumina reads was performed using the CLC Genomics Workbench v10.1.1 software (CLCInc, Aarhus, Denmark). After quality (quality score threshold 0.05) and length (above 40 nucleotides) trimming of the reads, contigs were generated by de novo assembly (CLC parameters: automatic determination of the word and bubble sizes with no scaffolding). The draft genome sequences of each strain were deposited at NCBI and annotated using the NCBI Prokaryotic Genome Annotation Pipeline. Statistics of all the ten draft genomes are presented in Table 1.

Assembly of the Nanopore reads was performed as follows. Fast5s from Nanopore sequencing were obtained with MinKNOW version 1.10.23 and were basecalled with ONT Albacore Sequencing Pipeline Software version 2.1.10 and reads passing the internal test were used for subsequent analysis. Porechop 0.2.1 (https://github.com/rrwick/Porechop) was used for adaptor trimming. Illumina paired-end reads were processed with trim_galore 0.4.0 (https://github.com/FelixKrueger/TrimGalore), to trim adaptor sequences. Nanopore reads were assembled using Canu 1.7 [48] with the “genomeSize = 5 m” and “minReadLength = 3000” options. For Nanopore-only assembly, one output contig was obtained, then polished three times using Pilon 1.22 (https://github.com/broadinstitute/pilon), with the “--mindepth 25” option. The contig was finally circularized using Circlator 1.5.1 (https://github.com/sanger-pathogens/circlator).

### 2.5. Genome Analysis

Phylogenetic and molecular evolutionary analyses were conducted using MEGA, version 7. An MLSA was performed using 13 concatenated housekeeping genes (*fusA, rpoD, acnA, purA, gyrB, recA, mdh, mtlD, groEL, secY, glyA, gapA, rplB*) retrieved from all the Pectobacterium *spp.* and Dickeya *spp*. strains to confirm their phylogenetic position within the reference strains *P. atrosepticum* ICMP1526^T^, *P. betavasculorum* NCPPB2795^T^, *P. parmentieri* RNS 08-42.1A^T^, *P. wasabiae* CFBP 3304^T^, *P. actinidiae* KKH3, *P. brasiliense* LMG21371^T^, *P. odoriferum* BCS7, *P. aroidearum* PC1, *D. dianthicola* NCPPB 453^T^, *D. dadantii* NCPPB 898^T^, and *D. solani* IPO2222^T^. The average nucleotide identity (ANI) value was calculated as previously proposed using the ANI calculator (http://enveomics.ce.gatech.edu/ani/, Atlanta, GA, USA). The in-silico DNA–DNA hybridization (*is*DDH) was evaluated using genome sequence-based species delineation (http://ggdc.dsmz.de/, Braunschweig, Germany) (Table 2).

The genome map of the *D. dianthicola* strain LAR.16.03.LID was generated using CGView Server [48]. Synteny analysis of the complete genomes of *D. dianthicola* LAR.16.03.LID, *D. dianthicola* ME23 and *D. dianthicola* RNS049 was performed using the MAUVE software [49]. Paired end reads for the strain LAR.16.03.LID were mapped against the two complete genome sequences of *D. dianthicola* strains ME23 and RNS049 with threshold set as 0.8 of identity on 0.5 of read length using CLC Genomics Workbench version 10.1.1 software. The mappings were used for detection of variations (SNPs and InDels) using the basic variant calling tool from the CLC genomic workbench version 10.1.1.

The presence of clustered regularly interspaced short palindromic repeats (CRISPRs) was determined using CRISPRfinder (http://crispr.i2bc.paris-saclay.fr/Server/, Orsay, France) [50]. The prophage identification tool PHAge Search Tool—Enhanced Release (PHASTER) was used to check for the regions containing prophage-like elements in bacterial genomes (http://phaster.ca/, Edmonton, AB, Canada) [51]. The Predicted resistome was checked using Resistance Gene Identifier tool (https://card.mcmaster.ca/analyze/rgi, Hamilton, ON, Canada). Finally, genomic regions containing secondary metabolite biosynthesis gene clusters were identified using the AntiSMASH server (version 4.1.0, https://doi.org/10.1093/nar/gkv437, Hørsholm, Denmark).

To investigate the phylogenetic position of the Moroccan *D. dianthicola* strain against the available genomes of this species in NCBI, an MLSA was generated using 15 housekeeping genes (*fusA, rpoD, leuS, rpoS, purA, infB, gyrB, recA, groEL, secY, glyA, gapA, rplB, dnaX, gyrA*) with the MEGA7 software.

### 2.6. Potato Tuber Rotting Assays

Bacterial strains from Morocco were cultivated in TY broth for 24 h at 28 °C in a rotary shaker set at 125 rpm. Bacterial cultures were washed twice, resuspended in 0.8% NaCl, and the optical density was adjusted to OD_600_ = 1.0. Potato tubers (cv. Bintje) were surface-disinfected by submerging them into a 5% sodium hypochlorite solution for 5 min. They were subsequently rinsed twice in distilled water and air dried at room temperature one day before inoculation. A total of 10 potato tubers were infected with 10 µL of cell suspension of each strain, along with 10 tubers with NaCl 0.8% alone as a negative control. After 5 days of incubation at 24 °C, the tubers were cut vertically through the inoculation points. Disease symptoms were evaluated to define five aggressiveness classes [52]. Significance of the observed differences was assessed using a Kruskal–Wallis test (*p* < 0.05).

## 3. Results

### 3.1. Diversity of the Pectinolytic Dickeya and Pectobacterium in Northern Morocco

From 2015 to 2017, our field inspections revealed the occurrence of blackleg symptoms in several potato growing areas, located in many townships distributed in three regions (Meknes, S1; Guigo, S2; Larache, S4) in northern Morocco (Figure 1). No symptoms were found in fields in the Boumia region (S3). Out of 200 strains isolated from plant symptoms, 140 provoked cavities on the pectate-containing medium. These were tested by PCR to evaluate whether they belonged to the Pectobacterium and Dickeya genera: 119 isolates generated amplification signals for either the Y1/Y2 or ADE1/ADE2 primer couples. Most of the isolates (83%) generated a signal with the Pectobacterium primers Y1/Y2, while the others (17%) did so with the Dickeya-specific ADE1/ADE2 primers.

All these PCR-positive Pectobacterium/Dickeya isolates were further characterized at species level based on their *gapA* gene sequence. Phylogenetic analyses using the Neighbor-Joining method (Mega7) of the *gapA* sequences are presented in the Appendix A. The regional diversity of the Dickeya and Pectobacterium isolates is summarized in Figure 1 (a detailed list is given in Appendix A). The samples of the Larache region (S4) showed the highest diversity of taxons with the presence of *D. dianthicola* (20 isolates), *P. polaris* (6 isolates)*, P. brasiliense* (5 isolates) and *P. carotovorum* (23 isolates). On the other hand, our investigations revealed the presence of only two species, *P. brasiliense* (53 isolates) and *P. carotovorum* (3 isolates), in the Meknes region (S1), and only a single one, *P. brasiliense* (9 isolates), in the Guigo region (S2).

Important information was also collected from the farmers regarding the potato variety, origin of the seed tubers (local production or importation), irrigation mode (surface water from a dam or underground water from wells) and geography (the sampled regions). We tested whether a correlation between these different parameters and the diversity of pathogens (the combination of Dickeya and Pectobacterium species) existed. Statistical analysis with the SAS (Statistical Analysis System, version 9.00, SAS Institute, 2002, Cary, NC, USA) software (Qui2 test with *p* < 0.05) revealed that the higher diversity of pathogens was associated with three confounding factors: geography (the unique Larache region), surface water irrigation and imported seed tubers (Appendix A).

### 3.2. Draft Genomes of 10 Pectinolytic Bacteria from Northern Morocco

A draft genome (Illumina technology) was used to consolidate the taxonomic position of 10 isolates belonging to the collected taxons. Between 705,755 and 17,016,482 trimmed reads were used for the contigs assembly of each of the 10 genomes. Characteristics of the draft genomes are presented in Table 1. Genome data were exploited to retrieve 13 housekeeping genes from each genome using BLAST. The concatenated genes were used for MLSA. The MLSA tree (Figure 2) showed a similar topology to the one generated by the *gapA* analysis (Appendix A).

Genomic data were also used to calculate ANI and *is*DDH values. Most of the Moroccan strains exhibited an ANI value higher than 95%, and an *is*DDH value higher than 70% with the closest type strains, confirming their taxonomic assignation. Strains belonging to the *P. versatile* clade showed an *is*DDH lower than 70%, but an ANI value higher than 95% with the strain SCC1. Recently, another study confirmed the classification of the strains S4.16.03.3I (= CFBP8660), S4.16.03.3F (= CFBP8659) and SCC1 into the *P. versatile* species [7].

### 3.3. Aggressiveness of the Pectinolytic Bacteria from Northern Morocco

Of the 10 Moroccan strains whose genome sequence is available (Table 1), all but 1 (*P. brasiliense* S4.16.03.1C) were tested for aggressiveness on potato tubers. *P. brasiliense* strain S4.16.03.1C was isolated from the same field as *P. brasiliense* strain S1.16.01.3K, and they showed 100% identify by ANI (Table 2). Hence, we retained one of the two for tuber maceration assays. For each of the nine strains, 10 tubers were inoculated, and the resulting maceration symptoms were classified into five symptomatic classes (Figure 3). In addition, 10 tubers were used as uninfected control. The aggressiveness was compared between all the strains using a Kruskal–Wallis test. All the pathogens provoked symptoms that are different to the control condition. No significant difference was observed between strains belonging to the same species. In contrast, *D. dianthicola* strains showed a higher aggressiveness when compared with Pectobacterium strains (*p* value < 0.05).

### 3.4. Complete Genome of D. dianthicola LAR.16.03.LID

The genome of *D. dianthicola* LAR.16.03.LID was the first complete genome of a *D. dianthicola* strain collected in Morocco, and the third *D. dianthicola* genome in the NCBI database that already hosted those of strains ME23 and RNS04.9 (Figure 4). Phylogenetic relationships between all *D. dianthicola* genomes available in NCBI were determined using MLSA. In the phylogenetic tree, the two Moroccan strains LAR.16.03.LID and S4.16.03.P2.4 appeared to be highly related. This could be explained by the close isolate locations, that were two potato fields separated only by a road. The two Moroccan *D. dianthicola* strains clustered with *D. dianthicola* strain ME23, which has been recently collected in a potato field in USA (Figure 4). The genomic relationship between *D. dianthicola* ME23 and *D. dianthicola* LAR.16.03.LID was confirmed using SNP/InDels calling. The SNP and InDel number in the LAR.16.03.LID genome reached 12,259 and 16,335, using the ME23 and *D. dianthicola* RNS04.9 genomes as a reference, respectively. The LAR.16.03.LID genome was annotated using the NCBI Prokaryotic Genome Annotation Pipeline. A graphical genome map is provided in Figure 5. The *D. dianthicola* LAR.16.03.LID genome exhibited a high synteny with that of *D. dianthicola* strains ME23 and RNS049, with the exception of some large insertion/deletions scattered in the genomes (Figure 6). These regions contained strain-specific genes with no counterpart in the other *D. dianthicola* genomes. In strains LAR.16.03LID and ME23, the analyses evidenced one strain-specific region that contained some mobile elements, such as genes from transposons and prophages; strain-specific regions 1 and 2 are presented in Figure 6. Additional information about these strain-specific regions are available in Appendix A. Strain RNS049 exhibited four strain-specific regions which also contain mobile elements (Figure 6 and Appendix A). Phaster analysis suggested the presence of intact prophages (Figure 6 and Appendix A). The CRISPR elements are very important features of bacterial genomes as they provide acquired immunity against viruses and plasmids [53]. The three *D. dianthicola* genomes hosted three or four CRISPR loci (Figure 6 and Appendix A).

The *D. dianthicola* LAR.16.03.LID genome exhibited an arsenal of virulence genes similar to that described for *D. dianthicola* RNS049 [39]. All the pectinase-encoding genes described in the model strain *D. dadantii* 3937 [54] were conserved in the three *D. dianthicola* genomes, with the noticeable exceptions of the lacking *pehK* gene (which encodes a predicted polygalacturonase) and the presence of a truncated form of the *pelA* gene (which encodes Pectate lyase A). Aside macerating enzymes determinants, other genes implicated in different stages of the host infection were conserved in the three *D. dianthicola* strains, including those involved in the resistance to oxidative stress, acidic pH (*cfa, asr*) and antimicrobial peptides (*arnB-T*, *sapABCDF*), synthesis of cell envelope components (such as *bscABCD* and *wza-wzb-wzc*), and siderophore synthesis and uptake (*acsF-A* and *cbrABCDE* for achromobactin and *fct-cbsCEBA* for chrysobactin) [54].

We strengthened this analysis using the resistance gene identifier (RGI) and AntiSMASH softwares. We searched for genes involved in the resistance to different families of antimicrobial compounds. No differences were observed between the three *D. dianthicola* genomes (Appendix A). The AntiSMASH analysis identified many secondary metabolite biogenesis clusters in *D. dianthicola* genomes that were already described in several Dickeya species, like those responsible for the synthesis of siderophores, cyanobactin with cytotoxic activity, bacteriocin, nonribosomal peptide-synthetase (NRPS) and arylpolyene (Figure 6 and Appendix A).

## 4. Discussion

The main objective of this study was to characterize the pectinolytic populations isolated from symptomatic potato plants in Morocco between 2015 and 2017. A set of 119 pectinolytic bacteria, belonging to the genus Pectobacterium or Dickeya, were isolated and characterized using the *gapA* gene marker in combination with MLSA and ANI. Most of the isolates (83%) belonged to the Pectobacterium genus: the *P. brasiliense* species dominated in the Meknes and Guigo regions, while *D. dianthicola* was identified in the Larache region only.

*P. brasiliense* caused major economic losses to several crops (potato, cucumber, paprika, etc.) in many countries, including Canada, USA, South Africa, China, Korea and New Zealand [55,56,57,58,59,60]. The wide host range of this pathogen could facilitate its survival even in harsh environments, by parasitizing many alternative host plants. In several studies, *P. brasiliense* isolates have been shown to be more aggressive than other Pectobacterium spp., except in the case of three Canadian strains exhibiting low aggressiveness [61]. In our study, the *P. brasiliense* isolates were as virulent as the other Pectobacterium strains. *P. brasiliense* was described in Morocco in 2012, and by now is the dominant species in two regions (S1 and S2). In the S2 region, the farmers use seed tubers produced in the S1 region, confirming the effective adaptation of this pathogen to the northern parts of Morocco.

The northern region (S4) exhibited the highest diversity of the pathogens. *P. versatile* and *P. Polaris*, described for the first time in Morocco, along with *P. brasiliense* and *D. dianthicola*, were isolated in the region S4. In this region, the majority of seed tubers were imported, and the irrigation water was derived from a dam. Either one or both agronomic practices could contribute to the wider diversity of pathogens in the Larache (S4) region than in the other investigated regions. While previous studies identified the *P. carotovorum* species as the most prevalent soft rot pathogens in Morocco [41,42], our study extended the diversity to other Pectobacterium and Dickyea species, including the recently described species *P. versatile* [7]. This species encompasses isolates, including the *P. versatile* strain SCC1 isolated in 1980 in Finland [62], which had been collected from a wide diversity of environments (host plants, surface waters) and geographic areas [7]. The presence of members of this clade in the potato field could be linked to irrigation, as *P. versatile* is also able to survive in this environment. This hypothesis remains to be investigated by sampling water from the dam. In addition, our study extended the worldwide distribution of this species to Morocco.

The international distribution of the genus Pectobacterium increases concerns about the economic losses caused by this bacterium to the potato growers. A study in the neighboring country Algeria, carried out between 2014 and 2015, revealed the presence of pectinolytic bacteria causing soft rot in potatoes that belonged to *P. brasiliense* and *P. carotovorum*, as judged by MLSA [63]. Ozturk et al. reported the presence of *P. atrosepticum*, *P. brasiliense*, *P. carotovorum* and *P. parmentieri* species in Turkey [64]. In Europe, the prevalence of different species belonging to the genera Pectobacterium and Dickeya detected in diseased potato plants differs from year to year and between countries, five bacterial species being the main causative agents of blackleg, namely *P. atrosepticum*, *P. parmentieri*, *P. brasilense*, *D. solani* and *D. dianthicola* [65].

The *D. dianthicola* species has been detected in Morocco in 2017 [44], and described as the main species causing losses in potato in North America [66]. More studies are needed for monitoring the spread of this highly aggressive pathogen. To reach this objective, we used Illumina and ONT sequencing technologies to assemble a complete genome of one *D. dianthicola* isolate that could be used, in the future, as a reference for studying the clonal variability of *D. dianthicola* populations in Morocco and elsewhere. Comparison of the three complete genomes available indicated the presence of several clusters that encode the biosynthesis of a number of secondary metabolites implicated in stress defense, possibly playing an important role during plant–bacteria interactions. For instance, bacteriocins are small molecules with bactericidal activity usually restricted to closely related species, increasing the competition during infection, while the production of arylpolyene, implicated in the protection against reactive oxygen species [67], has been recently described in *D. fangzhongdai* genomes [68].

## 5. Conclusions

This study revealed a wide diversity of Pectobacterium and Dickeya pathogens in northern Morocco, including *P. polaris* and *P. versatile*, that are reported for the first time in this country. In tuber maceration assays, the tested isolates of the emerging pathogen *D. dianthicola* were more aggressive than the Pectobacterium isolates. This feature should alert stakeholders to the threat that this pathogen poses to potato tuber production in northern Morocco. The nucletotide sequence data of the Dickeya and Pectobacterium Moroccan isolates, including a complete genome of *D. dianthicola*, 10 draft genomes and 119 partial sequences of the *gapA* gene, were deposited in a public database (NCBI GenBank) to be used as genetic resources for monitoring the spread of these pathogens in Northern Africa and elsewhere.

## Figures and Tables

**Figure 1 microorganisms-08-00895-f001:**
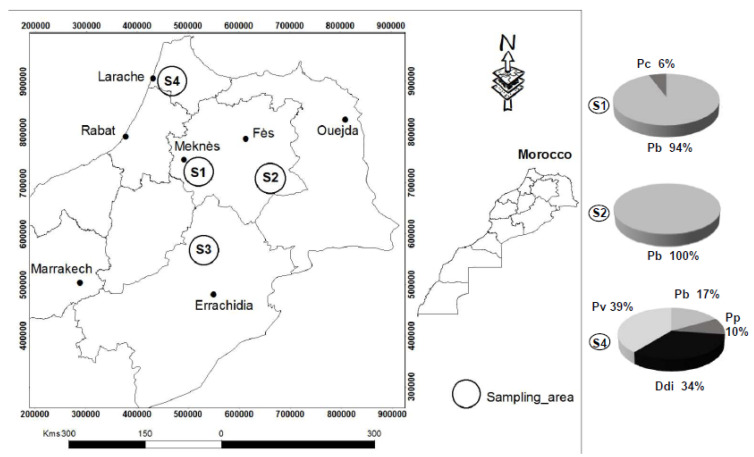
Species diversity of the Pectobacterium and Dickeya pathogens in the sampling areas of northern Morocco. Sampling of pathogens was performed in four potato growing areas in the northern Morocco: Meknes (S1), Guigo (S2), Boumia (S3) and Laraache (S4). The map of Morocco was generated by a free and open source geographic information system (https://www.qgis.org/fr/site/index.html). The pie charts represent diversity of the isolated pathogens. None of them were collected from the Boumia (S3) region. Legend: Ddi, *Dickeya dianthicola*; Pb, *Pectobacterium brasiliense*; Pc, *Pectobacterium carotovorum;* Pp, *Pectobacterium polaris*; Pv, *Pectobacterium versatile*.

**Figure 2 microorganisms-08-00895-f002:**
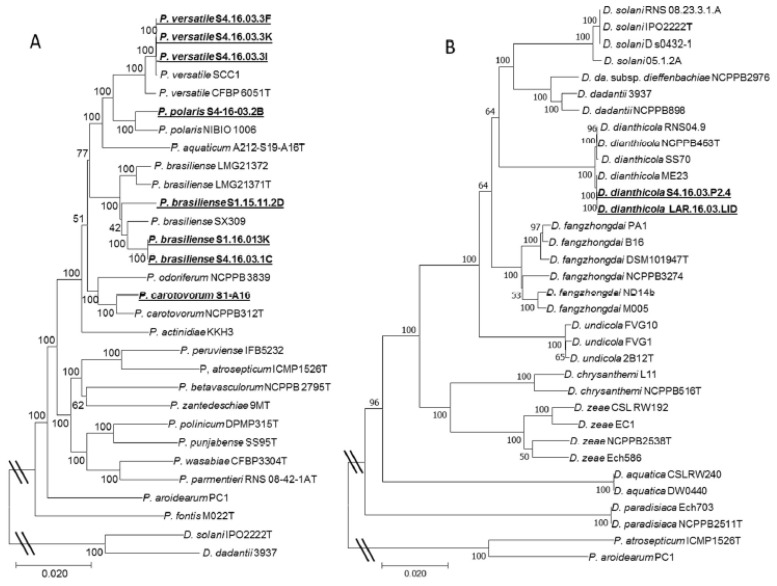
Phylogeny of the Moroccan strains based on MLSA. The phylogenetic trees were generated separately (**A**) for Pectobacterium and (**B**) for Dickeya strains. The alignment of the concatenated genes *fusA, rpoD, rpoS, acnA, purA, recA, mdh, mtlD, groEL, secY, glyA, gapA* and *rplB* was generated using ClustalW; the evolutionary history was inferred using the Neighbor-Joining method and the evolutionary distances were computed using the Maximum Composite Likelihood method. Phylogenetic analyses were conducted using MEGA7 software. The name of the Moroccan isolates is underlined.

**Figure 3 microorganisms-08-00895-f003:**
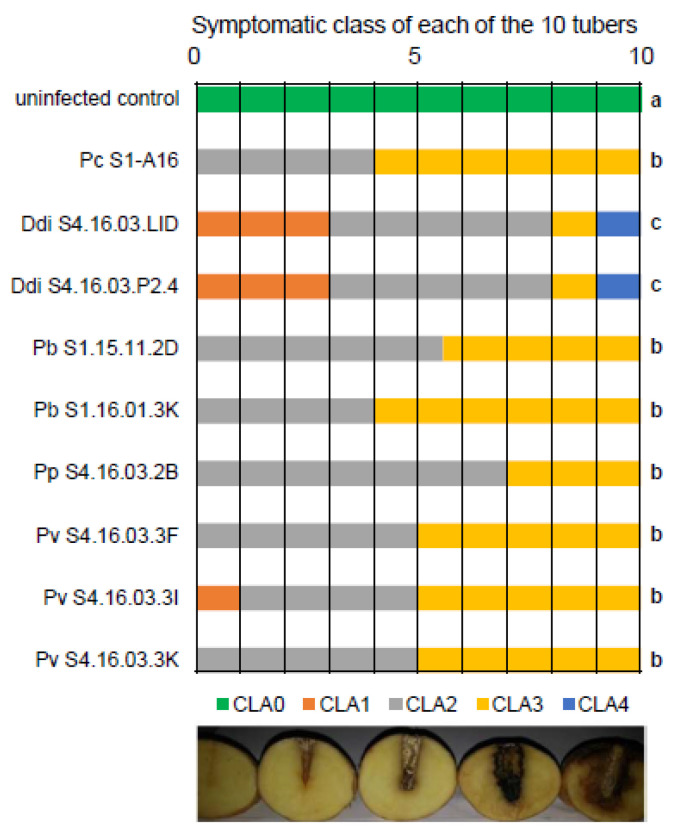
Virulence test of Pectobacterium and Dickeya on potato tubers. The symptoms provoked by each strain were compared by infecting 10 tubers per strain. The 10 non-inoculated tubers were used as control. Symptoms were classified into five classes (CLA0, CLA1, CLA2, CLA3, CLA4, according to increasing severity). The typology of these classes was illustrated by a picture of an example. Data were statistically analyzed by a Kruskal–Wallis test (α = 5%). Lower case letters on the right of the graph indicate statistical differences between the different inoculated pathogens.

**Figure 4 microorganisms-08-00895-f004:**
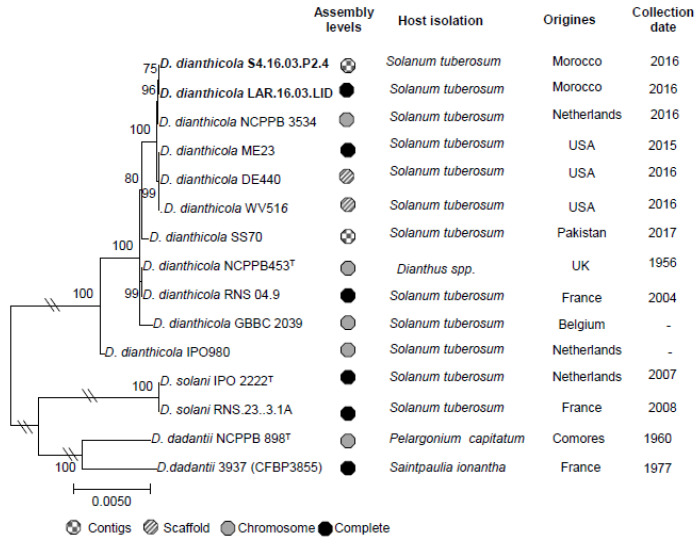
Phylogenetic analysis and characteristics of the Moroccan and NCBI Dickeya genomes. The genes *fusA, rpoD, leuS, rpoS, purA, infB, gyrB, recA, groEL, secY, glyA, gapA, rplB, dnaX* and *gyrA* were concatenated. The alignment was generated using ClustalW; the evolutionary history was inferred using the Neighbor-Joining method and the evolutionary distances were computed using the Maximum Composite Likelihood method. Phylogenetic analyses were conducted using MEGA7 software. The Moroccan *D. dianthicola* isolates are indicated in bold face.

**Figure 5 microorganisms-08-00895-f005:**
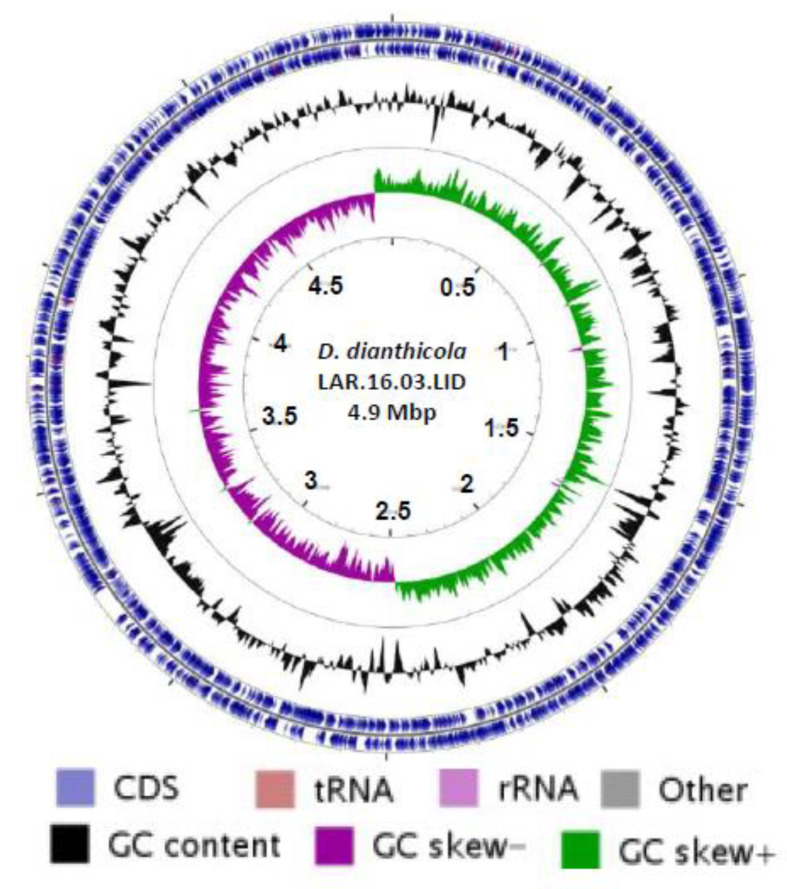
Circular map of the genome of *Dickeya dianthicola* LAR.16.03.LID. The genome size is 4,976,211 bp with 4223 predicted protein-coding genes. The GC content and GC skew are represented on the distance scale (in kbp) on the inner map. The arrows around the map indicate the deduced Coding DNA Sequences (CDS) and their orientation.

**Figure 6 microorganisms-08-00895-f006:**
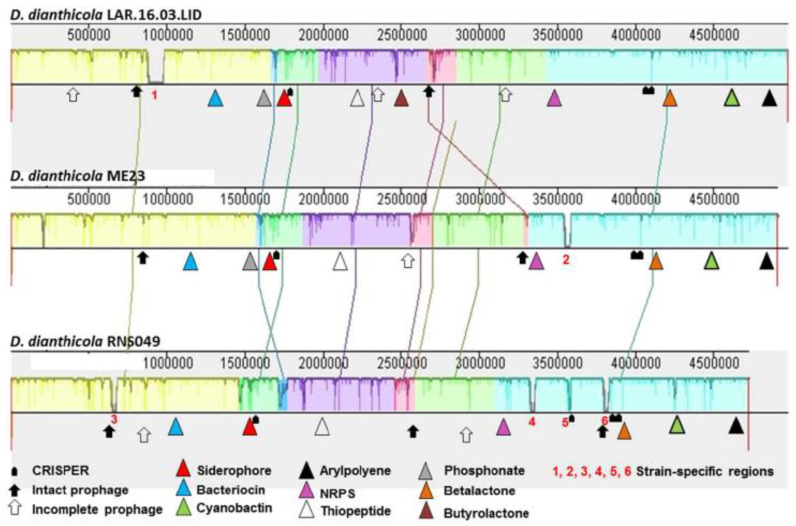
Synteny between the complete genomes of *D. dianthicola* LAR.16.03.LID, RNS049 and ME23 strains. Synteny analysis was performed using MAUVE software. The numbers indicate the position of strain specific genomic regions. The secondary pathway gene clusters were searched using AntiSMASH, the prophages were identified using PHASTER, and CRISPER loci were localized using CRISPER finder.

**Table 1 microorganisms-08-00895-t001:** Draft genome sequences of Pectobacterium and Dickeya strains isolated from Northern Morocco.

Organism	Accession Number	Genome Size	N50 (pb)	Number of Contigs	Coverage	Number of CDS	Number of tRNAs
*Pectobacterium polaris* S4.16.03.2B	QZDF00000000	4,862,009	155,865	65	41	4355	54
*Pectobacterium brasiliense* S1.16.01.3k	QZDG00000000	4,946,598	146,844	74	410	4337	36
*Pectobacterium brasiliense* S1.15.11.2D	QZDH00000000	4,818,836	99,392	91	420	4206	35
*Pectobacterium brasiliense* S4.16.03.1C	QZDI00000000	4,944,722	139,665	74	467	4336	37
*Pectobacterium carotovorum* S1-A16	QZDJ00000000	4,835,633	255,206	37	55	4261	63
*Pectobacterium versatile* S4.16.03.3I	QZDK00000000	4,854,084	8262	108	246	4247	34
*Pectobacterium versatile* S4.16.03.3K	QZDL00000000	4,870,940	90,195	106	237	4262	34
*Pectobacterium versatile* S4.16.03.3F	QZDM00000000	4,852,595	89,731	114	143	4244	40
*Dickeya dianthicola* S4.16.03.P2.4	QZDN00000000	4,865,147	92,028	101	415	4238	37
*Dickeya dianthicola* LAR.16.03.LID	QZDO00000000	4,863,939	71,707	108	344	4238	37

**Table 2 microorganisms-08-00895-t002:** Pairwise Average Nucleotide Identity (ANI) and in-silico DNA-DNA Hybridization (is-DDH) values of Pectobacterium and Dickeya strains isolated from Northern Morocco.

	ANI Values															
Strains	1	2	3	4	5	6	7	8	9	10	11	12	13	14	15	16
**1**-Pp NIBIO 1006^T^		**96.8**	92.9	92.9	93.5	93.7	93.5	93.7	93.4	93.4	93.5	93.5	92.2	79.3	79	79.1
**2**-Pp S4.16.03.2B	73.30		**93**	**92.9**	**93.5**	**93.7**	**93.5**	**93.7**	**93.4**	**93.5**	**93.5**	**93.4**	**92.2**	**78.9**	**79**	**79**
**3**-Pc ICMP5702^T^	52.3	52.10		97.2	92.6	92.9	92.8	92.8	95.1	95.2	95.2	95.2	94.8	78.9	78.9	78.9
**4-Pc S1-A16**	**52.2**	**51.90**	**76.30**		**92.6**	**92.9**	**92.7**	**92.9**	**95.3**	**95.3**	**95.3**	**95.3**	**94.8**	**78.8**	**79.6**	**79.6**
**5**-Pb LMG 21371^T^	54.4	54.5	50.9	50.9		96.1	95.9	96.1	92.2	92.2	92.2	92.3	93.00	78.6	79.5	79.5
**6-Pb S4.16.03.1C**	**56**	**55.7**	**52**	**51.7**	**68.5**		**96.3**	**100**	**92.3**	**91.2**	**91.1**	**91.2**	**91.9**	**78.7**	**79.6**	**79.6**
**7-Pb S1.15.11.2D**	**54.7**	**54.2**	**51.3**	**51**	**67.1**	**69.4**		**96.3**	**92.3**	**92.2**	**92.3**	**92.3**	**91.8**	**79.1**	**79.1**	**79.1**
**8-Pb S1.16.01.3K**	**56**	**55.7**	**76**	**51.7**	**68.5**	**100**	**71.4**		**92.4**	**92.4**	**92.4**	**92.3**	**91.9**	**79.6**	**79.6**	**79.7**
**9**-Pv SCC1	54.3	54.3	63.5	63.9	48.9	49.8	49.5	49.3		99.5	99.5	99.5	94.7	79.2	79.1	79.3
**10-Pv S4.16.03.3F**	**54.4**	**54.3**	**63.9**	**64.3**	**49.1**	**49.9**	**49.5**	**49.9**	**96.6**		**100**	**100**	**94.7**	**79.2**	**79.2**	**79.2**
**11-Pv S4.16.03.3k**	**54.4**	**54.3**	**63.8**	**64.2**	**49.2**	**49.9**	**49.5**	**49.9**	**96.6**	**99.3**		**100**	**94.7**	**79.3**	**79.3**	**79.4**
**12-Pv S4.16.03.3I**	**54.4**	**54.3**	**63.8**	**64.2**	**49.1**	**49.9**	**49.5**	**49.8**	**96.5**	**100**	**99.3**		**94.7**	**79.3**	**79.3**	**79.3**
**13**-Po BCS7	49.2	49	61.3	60.6	47.4	47.8	47.6	47.8	60.4	60.7	60.6	60.6		79.1	79.1	79.1
**14**-Ddi NCPPB 453^T^	21.1	20.7	20.5	20.7	20.9	21	20.6	21	21.2	21	21	21	21.1		**99.5**	**99.5**
**15-Ddi S4.16.03.P2.4**	**20.8**	**20.6**	**20.4**	**20.9**	**21.1**	**20.9**	**20.7**	**20.9**	**21**	**21**	**21**	**21**	**20.8**	**95.6**		**100**
**16-Ddi LAR.16.03.LID**	**20.8**	**20.6**	**20.4**	**21**	**21.1**	**20.9**	**20.6**	**20.9**	**21**	**21**	**21**	**21**	**20.8**	**95.6**	**100**	
	**is-DDH**															

Strains: **1**, *P. polaris* NIBIO1006^T^; **2**, *P. polaris* S4.16.03.2B **3**, *P. carotovorum* ICMP5702^T^; **4**, *P. carotovorum* S1-A16; **5**, *P. brasiliense* LMG21371^T^; **6**, *P. brasiliense* S4.16.03.1C; **7,**
*P. brasiliense* S1.15.11.2D **8**, *P. brasiliense* S1.16.01 3K; **9**, *P. versatile* SCC1; **10,**
*P. versatile* S4.16.03.3F; **11**, *P. versatile* S4.16.03.3k **12,**
*P. versatile* S4.16.03.3I; **13**, *P. odoriferum* BCS7; **14,**
*D. dianthicola* NCPPB 453^T^**, 15**, *D. dianthicola* S4.16.03.P2.4; **16**, *D. dianthicola* LAR.16.03.LID.

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
