# Peer review of "Diversity of Pectobacteriaceae Species in Potato Growing Regions in Northern Morocco"

_microorganisms, 2020, doi:10.3390/microorganisms8060895_

Round 1

Reviewer 1 Report

The manuscript of Oulghazi et al. is a thorough investigation of Pectobacteriaceae species present in potato growing districts in Northern Morocco. The manuscript is well written and clear, and the experiments are thoroughly described. I have some comments, one is the language where a hint of French order of words is present, such as water surface irrigation (instead of surface water irrigation, or irrigation with surface water), tubers seeds (instead of seed tubers) to mention two. I think language edition would be beneficial. My second concern is the quality of the figures, their resolution appears low, or is it just a problem of this PDF? Especially the Figure 1 seems to have low resolution. Thirdly, I think a control should be present in the Figure 3. This would also take care of the situation with the missing a from the statistical differences, because it feels strange that they are marked b and c, but there is no a. I suspect that was the control. I would also like to suggest a shorter title for the manuscript, for example “Diversity of Pectobacteriaceae species in potato growing regions in Northern Morocco”. Minor comments concern the sentence on lines 116-117, that could be written as follows: Ten strains representing five species identified with gapA sequencing … Also, line 318 is interrupted. I also wonder why the supplementary tables were not visible.

  What do you want to do ? New mailCopy   What do you want to do ? New mailCopy

Author Response

We thank the reviewers for their comments and suggestions to improve the manuscript. Our main modifications are in blue color in the revised text.

Reviewer1 : The manuscript of Oulghazi et al. is a thorough investigation of Pectobacteriaceae species present in potato growing districts in Northern Morocco.

Point 1. The manuscript is well written and clear, and the experiments are thoroughly described. I have some comments, one is the language where a hint of French order of words is present, such as water surface irrigation (instead of surface water irrigation, or irrigation with surface water), tubers seeds (instead of seed tubers) to mention two. I think language edition would be beneficial. 

Response 1. A language editing has been performed by two English-speaking colleagues.

Point 2. My second concern is the quality of the figures, their resolution appears low, or is it just a problem of this PDF? Especially the Figure 1 seems to have low resolution.

Response 2. We used high quality pdf. We hope that the novel version will be uploaded properly. In addition, according the reviewer2’s comments, we modified the figure 1.

Point 3. Thirdly, I think a control should be present in the Figure 3. This would also take care of the situation with the missing a from the statistical differences, because it feels strange that they are marked b and c, but there is no a. I suspect that was the control.

Response 3. There is an infected control at the top of the figure (with CLA0 class symptoms in green): this condition is statistically different from all the others; this was indicated by the letter ‘a’ on the right.

Point 4: I would also like to suggest a shorter title for the manuscript, for example “Diversity of Pectobacteriaceae species in potato growing regions in Northern Morocco”.

Response 4: the title was changed in the revised version.

Point 5. Minor comments concern the sentence on lines 116-117, that could be written as follows: Ten strains representing five species identified with gapA sequencing … Also, line 318 is interrupted. Response 5: These changes were done.

Point 6. I also wonder why the supplementary tables were not visible. 

Response 6: In the revised manuscript, we will ensure an access to supplementary tables.

Reviewer 2 Report

The manuscript titled “A Wide Diversity of Pectinolytic Pathogen Emerged From  Different Potato Growing Areas of Northern Morocco” is devoted to investigation of species diversity for pectolytic potato pathogens in Morocco. Samplings were conducted in three major potato growing areas over three years (2015-2017). Pathogens were characterized by sequencing the gapA gene marker and genomes using Illumina and Oxford Nanopore technologies. 119 pathogenic isolates were identified to species level, including  P. versatile, P. carotovorum, P. polaris, P. brasiliense and D. dianthicola. The taxonomic assignation  was confirmed by draft genome analyses of 10 representatives belonging to the collected species. This study is important for understanding distribution of some Dickeya and Pectobacterium species in Morocco and for further epidemiological studies.

There are some minor corrections must be done before the publication:

Line 61

  1. D.solani emerged in potato cultivars in the 2000s in several European countries, and Israel [22].

- In fact, D. solani has been found in Russia, Turkey, Georgia, Brazil as well (references below). Please, re-write  

Kabanova AP, Shneider MM, Korzhenkov AA, Bugaeva EN, Miroshnikov KK, Zdorovenko EL, Kulikov EE, Toschakov SV, Ignatov AN, Knirel YA, Miroshnikov KA. Host specificity of the Dickeya bacteriophage PP35 is directed by a tail spike interaction with bacterial O-antigen, enabling the infection of alternative non-pathogenic bacterial host. Frontiers in microbiology. 2019 Jan 11;9:3288.

Ozturk M, Aksoy HM. First report of Dickeya solani associated with potato blackleg and soft rot in Turkey. Journal of Plant Pathology. 2017;99(1).

Cardoza YF, Duarte V, Lopes CA. First report of blackleg of potato caused by Dickeya solani in Brazil. Plant Disease. 2017 Jan 23;101(1):243-.

Tsror L, Erlich O, Lebiush S, Wolf J van der, Czajkowski R, Mozes G, Sikharulidze Z, Daniel B B, 2011. First report of potato blackleg caused by a biovar 3 Dickeya sp. in Georgia. New Disease Reports. Article 1. http://www.ndrs.org.uk/pdfs/023/NDR_023001.pdf

Line 119

The quantity and quality control of the DNA was completed using a NanoDrop device and agarose gel electrophoresis at 1.0 %.

Suggested: … 1% agarose gel electrophoresis.

Line 198

Ten potato tubers were infected with 10 μl of cell suspension of each strain, along with 10 tubers with NaCl 0.8% alone as a negative control. After 5 days of incubation at 24 °C, the tubers were cut vertically through the inoculation points and classified into five aggressiveness grades.

Please, describe the method of inoculation or give the method source and describe rating scale and link it to the Figure 3B.

Line 216

The regional diversity of the Dickeya and Pectobacterium isolates is summarized in Figure1 (a detailed list is given in TableS2).

So as the assay was made in 3 regions  (Larache region, Meknes region and  Guigo region) there is no need to show the map of the whole country (Fig. 1), The targeted regions look too small for clear understanding of their location.

Author Response

We thank the reviewers for their comments and suggestions to improve the manuscript. Our main modifications are in blue color in the revised text.

Reviewer2 : The manuscript titled “A Wide Diversity of Pectinolytic Pathogen Emerged From  Different Potato Growing Areas of Northern Morocco” is devoted to investigation of species diversity for pectolytic potato pathogens in Morocco. Samplings were conducted in three major potato growing areas over three years (2015-2017). Pathogens were characterized by sequencing the gapA gene marker and genomes using Illumina and Oxford Nanopore technologies. 119 pathogenic isolates were identified to species level, including  P. versatile, P. carotovorum, P. polaris, P. brasiliense and D. dianthicola. The taxonomic assignation  was confirmed by draft genome analyses of 10 representatives belonging to the collected species. This study is important for understanding distribution of some Dickeya and Pectobacterium species in Morocco and for further epidemiological studies. There are some minor corrections must be done before the publication:

Point 1. Line 61

D.solani emerged in potato cultivars in the 2000s in several European countries, and Israel [22].

- In fact, D. solani has been found in Russia, Turkey, Georgia, Brazil as well (references below). Please, re-write  

- Kabanova AP, Shneider MM, Korzhenkov AA, Bugaeva EN, Miroshnikov KK, Zdorovenko EL, Kulikov EE, Toschakov SV, Ignatov AN, Knirel YA, Miroshnikov KA. Host specificity of the Dickeya bacteriophage PP35 is directed by a tail spike interaction with bacterial O-antigen, enabling the infection of alternative non-pathogenic bacterial host. Frontiers in microbiology. 2019 Jan 11;9:3288.

- Ozturk M, Aksoy HM. First report of Dickeya solani associated with potato blackleg and soft rot in Turkey. Journal of Plant Pathology. 2017;99(1).

- Cardoza YF, Duarte V, Lopes CA. First report of blackleg of potato caused by Dickeya solani in Brazil. Plant Disease. 2017 Jan 23;101(1):243-.

- Tsror L, Erlich O, Lebiush S, Wolf J van der, Czajkowski R, Mozes G, Sikharulidze Z, Daniel B B, 2011. First report of potato blackleg caused by a biovar 3 Dickeya sp. in Georgia. New Disease Reports. Article 1. http://www.ndrs.org.uk/pdfs/023/NDR_023001.pdf

Response 1: we added the above mentioned, suggested references.

Point 2. Line 119

The quantity and quality control of the DNA was completed using a NanoDrop device and agarose gel electrophoresis at 1.0 %.

Suggested: … 1% agarose gel electrophoresis.

Response 2: the sentence is revised as requested.

Point3. Line 198

Ten potato tubers were infected with 10 μl of cell suspension of each strain, along with 10 tubers with NaCl 0.8% alone as a negative control. After 5 days of incubation at 24 °C, the tubers were cut vertically through the inoculation points and classified into five aggressiveness grades.

Please, describe the method of inoculation or give the method source and describe rating scale and link it to the Figure 3B.

Response 3: We added a reference for this tuber maceration assays that we used in a previous paper to compare aggressiveness of Dickeya and Pectobacterium strains. In addition, we modified figure 3 to improve its clarity.

Point4: Line 216

The regional diversity of the Dickeya and Pectobacterium isolates is summarized in Figure1 (a detailed list is given in TableS2). So as the assay was made in 3 regions  (Larache region, Meknes region and  Guigo region) there is no need to show the map of the whole country (Fig. 1), The targeted regions look too small for clear understanding of their location.

Response 4. A novel figure 1 is proposed in the revised manuscript with a map centered on northern Morocco.

Round 2

Reviewer 1 Report

  What do you want to do ? New mailCopy   What do you want to do ? New mailCopy